# Negative mpMRI Rules Out Extra-Prostatic Extension in Prostate Cancer before Robot-Assisted Radical Prostatectomy

**DOI:** 10.3390/diagnostics12051057

**Published:** 2022-04-23

**Authors:** Eoin Dinneen, Clare Allen, Tom Strange, Daniel Heffernan-Ho, Jelena Banjeglav, Jamie Lindsay, John-Patrick Mulligan, Tim Briggs, Senthil Nathan, Ashwin Sridhar, Jack Grierson, Aiman Haider, Christos Panayi, Dominic Patel, Alex Freeman, Jonathan Aning, Raj Persad, Imran Ahmad, Lorenzo Dutto, Neil Oakley, Alessandro Ambrosi, Tom Parry, Veeru Kasivisvanathan, Francesco Giganti, Greg Shaw, Shonit Punwani

**Affiliations:** 1Division of Surgery & Interventional Science, University College London, Charles Bell House, 3rd Floor, 43-45 Foley Street, London W1W 7TS, UK; j.grierson@ucl.ac.uk (J.G.); veeru.kasi@ucl.ac.uk (V.K.); f.giganti@ucl.ac.uk (F.G.); gregshaw@nhs.net (G.S.); 2Department of Urology, University College Hospital London, Westmoreland Street Hospital, 16-18 Westmoreland Street, London W1G 8PH, UK; jelena.banjeglav@nhs.net (J.B.); jamie.lindsay@nhs.net (J.L.); john-patrick.mulligan@doctors.org.uk (J.-P.M.); tim.briggs@nhs.net (T.B.); s.nathan@nhs.net (S.N.); ashwin.sridhar@nhs.net (A.S.); 3Department of Radiology, University College London Hospitals, 235 Euston Road, London NW1 2BU, UK; clare.allen1@nhs.net (C.A.); tomstrange@doctors.net.uk (T.S.); d.heffernanho@nhs.net (D.H.-H.); shonit.punwani@gmail.com (S.P.); 4Department of Histopathology, University College Hospital London, 235 Euston Road, London NW1 2BU, UK; aiman.haider@nhs.net (A.H.); christos.panayi@nhs.net (C.P.); dominic.patel@ucl.ac.uk (D.P.); alex.freeman2@nhs.net (A.F.); 5North Bristol Hospitals Trust, Department of Urology, Southmead Hospital, Southmead Lane, Westbury-on-Trym, Bristol BS10 5NB, UK; jonathan.aning@nbt.nhs.uk (J.A.); rajpersad@bristolurology.com (R.P.); 6Department of Urology, Queen Elizabeth University Hospital, NHS Greater Glasgow & Clyde, 1345 Govan Road, Glasgow G51 4TF, UK; imran.ahmad@glasgow.ac.uk (I.A.); lor.dutto@gmail.com (L.D.); 7Department of Urology, Sheffield Teaching Hospitals NHS Trust, Royal Hallamshire Hospital, Glossop Road, Sheffield S10 2JF, UK; neil.oakley@nhs.net; 8Faculty of Medicine and Surgery, Vita-Salute San Raffaele University, 20132 Milano, Italy; ambrosi.alessandro@hsr.it; 9Centre for Medical Imaging, University College London, Charles Bell House, 2nd Floor, 43-45 Foley Street, London W1W 7TS, UK; thomas.parry.19@ucl.ac.uk

**Keywords:** extra-prostatic extension, magnetic resonance imaging, radical prostatectomy, nerve-sparing, prostate cancer, staging, diagnostic accuracy

## Abstract

*Background:* The accuracy of multi-parametric MRI (mpMRI) in the pre-operative staging of prostate cancer (PCa) remains controversial. *Objective:* The purpose of this study was to evaluate the ability of mpMRI to accurately predict PCa extra-prostatic extension (EPE) on a side-specific basis using a risk-stratified 5-point Likert scale. This study also aimed to assess the influence of mpMRI scan quality on diagnostic accuracy. *Patients and Methods:* We included 124 men who underwent robot-assisted RP (RARP) as part of the NeuroSAFE PROOF study at our centre. Three radiologists retrospectively reviewed mpMRI blinded to RP pathology and assigned a Likert score (1–5) for EPE on each side of the prostate. Each scan was also ascribed a Prostate Imaging Quality (PI-QUAL) score for assessing the quality of the mpMRI scan, where 1 represents the poorest and 5 represents the best diagnostic quality. *Outcome measurements and statistical analyses:* Diagnostic performance is presented for the binary classification of EPE, including 95% confidence intervals and the area under the receiver operating characteristic curve (AUC). *Results:* A total of 231 lobes from 121 men (mean age 56.9 years) were evaluated. Of these, 39 men (32.2%), or 43 lobes (18.6%), had EPE. A Likert score ≥3 had a sensitivity (SE), specificity (SP), NPV, and PPV of 90.4%, 52.3%, 96%, and 29.9%, respectively, and the AUC was 0.82 (95% CI: 0.77–0.86). The AUC was 0.76 (95% CI: 0.64–0.88), 0.78 (0.72–0.84), and 0.92 (0.88–0.96) for biparametric scans, PI-QUAL 1–3, and PI-QUAL 4–5 scans, respectively. *Conclusions:* MRI can be used effectively by genitourinary radiologists to rule out EPE and help inform surgical planning for men undergoing RARP. EPE prediction was more reliable when the MRI scan was (a) multi-parametric and (b) of a higher image quality according to the PI-QUAL scoring system.

## 1. Introduction

Multi-parametric magnetic resonance imaging (mpMRI) of the prostate is a valuable tool in the prostate cancer (PCa) diagnostic pathway [1]. Once a diagnosis of PCa has been made on biopsy, the correct assessment of the tumour stage, particularly identifying pathological extra-prostatic extension (EPE), is crucial for directing correct surgical planning, including nerve-sparing (NS), bladder neck sparing, and the extent of apical dissection [2,3,4]. These decisions are pivotal in promoting optimal post-operative functional outcomes without compromising cure from PCa [5]. Moreover, the EPE of a tumour is associated with a higher risk of positive surgical margins (PSM), biochemical recurrence, metastatic disease, and cancer-specific mortality [6,7,8].

It is well recognised that prostate mpMRI has good concordance with PCa on the histopathological evaluation of the final RP specimen for lesions that are within the gland [9]. However, the accurate assessment of PCa EPE has proven to be a more challenging domain for all pre-operative radiological imaging modalities, including MRI. In 2016, de Rooij et al. reported their diagnostic meta-analysis on the accuracy of mpMRI for the local staging of PCa [10]. Overall, MRI had a sensitivity (SE) of 55% (95% CI: 47–63%) and a specificity (SP) of 91% (95% CI: 88–93%) for the prediction of stage pT3a disease, though the authors noted considerable heterogeneity in the studies included. Since this review, although use of mpMRI has become vastly more prominent in the PCa diagnostic pathway around the world, a recent review by Abrams-Pompe et al. clearly demonstrates the ongoing challenges of using this modality for staging purposes [11]. In their review, they continued to demonstrate enormous variation in the diagnostic performance with SE ranging from 0–100%, which the authors attribute to differing sample sizes, event rates, reader experience levels, study designs, heterogeneous non-standardized reporting of mpMRI, and even different definitions of histopathological EPE. Interestingly, although clearly a critical part of the performance and interpretation of mpMRI, technical scan quality has never been assessed as a contributing factor to the accuracy of mpMRI for PCa staging, perhaps because until recently there has been little attention to grading mechanisms for scan quality [12,13,14]. Thus, although mpMRI is now routinely used for pre-operative PCa staging, variation in its performance is still huge and, as such, its role in surgical planning remains controversial.

The objective of this study was to assess the accuracy of mpMRI for the pre-operative local staging of PCa for men undergoing RARP at a high-volume UK academic centre whose scans were performed within the regional referral network. A key secondary objective was to evaluate the effect of technical scan quality on the diagnostic accuracy of EPE prediction.

## 2. Materials and Methods

### 2.1. Study Design and Patient Population

We undertook a retrospective review of mpMRI performed in men who had already undergone robot-assisted RP (RARP) at our centre as part of the NeuroSAFE PROOF trial (NCT03317990, ethics approval 17/LO/1978; see Appendix A for full inclusion/exclusion criteria). The STROBE checklist for the reporting of observational studies is included as an online Appendix A. Though all the men included in this study were participants in the NeuroSAFE PROOF trial, mpMRI scans were performed as part of their routine clinical care. All participants provided written informed consent. 

### 2.2. Standard of Reference

RARP was performed by three experienced surgeons (GS, SN, TB). Histopathologic examination was performed by a consultant genitourinary pathologist according to the International Society of Urological Pathology (ISUP) 2014 consensus statement [15,16]. Pathologic EPE was diagnosed separately for each side of the prostate where cancer cells were seen outside of the prostate capsule or pseudo-capsule. Seminal vesicle invasion (pT3b) on its own was not classified as EPE. Where an intraprostatic PSM was observed with no evidence of ipsilateral EPE elsewhere, the lobe was excluded from analysis as it is not possible to know whether this focus of PCa extended beyond the limit of the prostate. 

### 2.3. Imaging Protocols and Evaluation

Standard of clinical care mpMRI were performed at 13 different locations in the north and east of London as part of the regional NHS hub and spoke model for the referral of PCa cases for surgery at our centre (Appendix A). mpMRI was performed without endorectal coil. Scans were performed using a variety of magnets (1.5 T and 3 T) from different vendors (Philips, GE, and Siemens). 

All mpMRI scans were reviewed independently by 3 genitourinary radiologists (CA, DH, TS) who were blinded to the RP final pathology results and who reviewed images under invigilated circumstances in the same room at the same time. Reporting was performed on a bespoke MRI EPE case reporting form (Appendix A). Images were reviewed and reported in a locked sequential manner such that T2WI was reported first, then DWI (including apparent diffusion coefficient (ADC) reconstruction maps), then DCE, then the blood serum prostate specific antigen (PSA) level was provided, and finally prostate biopsy information was provided. Where DCE sequences were not available for a given scan (biparametric scans), the score for DCE specifically was omitted by radiologists, but scores were still provided after PSA and biopsy information. At each point, the 3 radiologists separately rated the likelihood of EPE using a risk-stratified 5-point Likert scale: 1, highly unlikely; 2, unlikely; 3, equivocal or indeterminate; 4, likely; and 5, highly likely. Once a score had been assigned and the next imaging sequence or information was provided, the radiologist was not allowed to change any preceding score. Radiologists scored for EPE specifically, but not seminal vesicle invasion (pT3b).

The 3 radiologists (CA, DH, TS) were all fellowship trained and had 20, 5, and 1 years of experience as a consultant, reading >3000, >750, and >750 prostate MRI scans per year, respectively. Before initiating interpretation, the radiologists attended an in-person training session where examples of the MRI scans of patients with and without EPE were reviewed alongside whole mount pathological images from the RP specimen. 

Separate to the process described above, the quality of all scans was assessed by means of the Prostate Imaging Quality (PI-QUAL) score from 1 to 5, where 1 represents the poorest and 5 represents the best quality scan (Appendix A), by two radiologists (CA and FG) [12]. The PI-QUAL score is derived by evaluating mpMRI scans against a defined set of objective quality criteria in line with Prostate Imaging-Reporting and Data System (PI-RADS) v2 guidelines [17] that cover the adequacy of all three of the mpMRI sequences and additional objective criteria considered to be consistent with a high-quality scan. PI-QUAL scores were given blinded to the EPE Likert score and RP pathology after a wash-out period >4 months. In the case of discordance, the PI-QUAL score was discussed until consensus was reached. 

### 2.4. Sample Size and Power Calculation

Based on the performance reported in the meta-analysis by Rooij et al. [10], the 217 prostate lobes reviewed by the three readers within this study provide 90% power to detect a sensitivity of 80% (95% CI: 65–95) and a specificity of 90% (95% CI: 80–100), assuming a 19% EPE prevalence and a 10% loss to follow-up. The sample size and power were calculated in Stata 16 (College Station, TX, USA). 

### 2.5. Statistical Analysis

First, descriptive statistics were used to compare patients with and without EPE. Continuous variables were reported as means and compared by 2-sided independent t tests, whereas categorical data were reported as proportions and were tested using the chi-squared test. For assessing diagnostic accuracy of mpMRI, the prostate lobe was the unit of analysis as treatment planning incorporating mpMRI occurs on a lobe basis. A Likert score ≥3 was considered positive for EPE. Diagnostic accuracy measures (SE, SP, NPV, PPV, and AUC) were calculated for each reader and combined. We based statistical significance on 95% CIs derived from the exact binomial method. All analyses were conducted in SPSS (version 27, IBM).

## 3. Results

A total of 1125 men underwent RARP at our centre between June 2018 and December 2019. Of these, 127 were participants in the NeuroSAFE PROOF study and eligible for inclusion. Four men were excluded because their mpMRI was not available for review. Of the remaining 123 men, 36 (29.3%) had PSM, of which 13 (10.6%) were ≤1 mm, 3 (2.4%) were 1–3 mm, and 20 (16.3%) were >3 mm or multifocal. Of the 246 lobes available for the purposes of evaluating whether EPE was present or not, a further 15 lobes were excluded because of intra-prostatic PSM (Figure 1), meaning that 121 men and 231 lobes were included in the final analysis. Representative mpMRI and matched patient RP pathology images can be seen in Figure 2.

Clinical, radiological, and final pathological characteristics of the patient cohort are shown in Table 1. Mean age was 56.9 years, mean PSA level was 8.9 ng/mL, and mean prostate volume on mpMRI was 36.6 cc ± 15. On final pathology, 82 of 121 (67.8%) men had organ-confined disease (pT2a-pT2c). EPE was seen in 39 of 121 men (32.2%), including in all seven men who had pT3b disease, and 43 of 231 lobes (18.6%). In 35 of 39 men (89.7%), EPE was unilateral. Preoperative PSA, clinical (i.e., biopsy) ISUP score, EAU risk category, DRE, RP specimen tumour volume, and RP ISUP score differed significantly between the groups with EPE and without EPE (all *p* < 0.05).

A total of 112 of the scans used for EPE assessment were original pre-biopsy mpMRI, whilst nine men had mpMRI repeated after their prostate biopsy (Appendix A). Of these nine men, the mean number of days from biopsy to repeat scan was 163 (range; 63–475). All nine men had their repeat mpMRI scan performed at the academic centre and had a mean of 13 days (range; 2–25) from the repeat scan until RARP. 

### 3.1. mpMRI Prediction of EPE

The diagnostic accuracy of mpMRI for EPE is presented in Table 2. The detection rates of pathological EPE according to each Likert score are presented in Figure 3 and tabulated in Appendix A. For Likert ≥ 3 on final read (i.e., once all imaging sequences, PSA, and biopsy information was available), there was an SE of 90.4% (CI: 83.8–94.9), SP of 52.3% (48–56.5), PPV of 29.9% (25.3–34.8), and NPV of 96% (93.2–97.9). When analysis was instead performed on a per patient level, for Likert ≥ 3 on final read there was an SE of 97.4%, SP of 24.4%, PPV of 38% and NPV of 95.2% (see Appendix A). Inter-observer agreement was determined using a weighted Kappa coefficient and is provided in Appendix A, showing substantial agreement between all readers on the final Likert score for the detection of EPE.

### 3.2. Scan Quality and Diagnostic Performance

Eighteen scans were biparametric and therefore could not be assigned a PI-QUAL score (Figure 1). A total of 103 scans were scored according to PI-QUAL; 63 and 40 scans were classified with scores of 1–3 and 4–5, respectively. Diagnostic accuracy is presented in Table 3 and Appendix A, showing that biparametric scans performed the least well, followed by PI-QUAL 1–3 scans and PI-QUAL 4–5 scans, which were best interpreted. AUCs were 0.76 (95% CI: 0.64–0.88), 0.78 (0.72–0.84), and 0.92 (0.88–0.96) for biparametric scans, PI-QUAL 1–3, and PI-QUAL 4–5 scans, respectively. PI-QUAL 4–5 scans had an SE of 100% (90.3–100) and an SP of 57% (49.7–64.1).

### 3.3. Accuracy of EPE with Additional Sequences and Clinical Information

Diagnostic accuracy improved with the addition of each imaging sequence and then further with the provision of PSA and biopsy information. The AUCs for T2WI alone, +DWI, +DCE, +PSA, and final (i.e., +biopsy information), were 0.62 (95% CI: 0.56–0.69), 0.69 (0.64–0.75), 0.76 (0.71–0.82), 0.79 (0.74–0.84), and 0.82 (0.77–0.86), respectively (see Appendix A for ROC curve).

## 4. Discussion

The accurate and precise prediction of EPE will result in better surgical outcomes by guiding surgeons to achieve the often-competing surgical goals of minimising PSM (by avoiding EPE) whilst maximising safe NS. Our data on EPE prediction by genitourinary radiologists using a Likert scale to attribute EPE status demonstrate the excellent ability of mpMRI to rule out EPE (SE of 90.4%, NPV of 96%). 

Our study is the first to demonstrate improved EPE prediction with a higher objectively scored mpMRI scan quality (PI-QUAL). This finding strongly supports the concept that a high SE is dependent on good quality mpMRI scans and, we believe, may explain the poor or heterogeneous correlation of mpMRI and EPE status in previous studies. 

The high SE for the prediction of EPE reported in this manuscript (90.4%) appears to come at the cost of a relatively lower SP (52.3%). Arguably, with the primary aim of optimising oncological outcomes by reducing PSM, the interpretation of mpMRI findings should be shifted towards obtaining a high SE in order to accurately rule out EPE, and thus, when a decision for NS is made, this is likely to be appropriate. On the other hand, with a low SP and PPV, if NS is not performed based solely on the mpMRI Likert score (3–5), in this study many men would not receive NS in whom it may have safely been performed, thus potentially negatively and unnecessarily affecting functional outcomes. 

These results, a high SE and a lower SP, are at odds with the findings of the systematic review and meta-analysis of MRI for local PCa staging performed by de Rooij et al. in 2016. This review included 75 studies and demonstrated a high SP (91%) with a poor and heterogeneous SE (51%) [10]. Since this review, huge interest has continued in this topic as highlighted by the review by Abrams-Pompe et al., who included a further 48 studies that had been published with a significant ongoing variation in diagnostic accuracy reported [11]. For instance, Davis et al. reported an SE of 13% in their 2016 study of 133 men using scans performed at community radiology centres [18], whereas Alessi et al. reported an SE of 99% in 2019 in a study of 301 men where scans were performed at a single academic centre and reported according to the European Society of Urogenital Radiology (ESUR) PI-RADS EPE score version 2 [19]. In the UK, 98% of hospital trusts now perform pre-biopsy mpMRI for men with suspicion of PCa diagnosis [20]. In 2004, prior to the widespread dissemination and prominence of mpMRI in the diagnostic pathway in the UK, Allen et al. reported on 56 patients and found an SE of 50–72% and an SP of 84–86% for MRI EPE prediction [21].

Possible explanations for this broad variation in the accurate prediction of EPE published in the literature include differences in study design, measurement metrics, definitions of EPE, and clinical practice. In terms of different clinical practices, scans are performed in different settings and on different patient populations with different EPE prevalence. Additionally, scans may be performed before and after biopsy, using different scanners (magnets and manufacturers) and read by radiologists of differing experience. In terms of study design, there are different (and non-standardised) methods for reporting EPE where clinical information may or may not be provided to readers and where analysis can be undertaken in a variety of ways. For example, in our study performing analysis ‘per prostate lobe’ (Table 2) or ‘per patient’ (Appendix A) slightly increased SE, but considerably reduced SP and AUC overall (0.82 to 0.61). Such findings are consistent with the pooled sensitivity analysis performed by de Rooij et al., who demonstrated that the prediction of EPE was more accurate on a ‘region only’ compared to a ‘patient only’ level [10]. Moreover, different patient populations can have a marked effect on diagnostic accuracy. For instance, Falagario et al. demonstrated an AUC for EPE prediction using mpMRI of 0.73 for patients with high-risk PCa compared to 0.52 for patients with low- or very low-risk disease [22]. In the present study, 71.9% of patients had EAU high-risk PCa, which may help to also explain the strong diagnostic accuracy results (AUC of 0.82). The study by Falagario and colleagues may also demonstrate the importance of the definition of EPE on the diagnostic performance of mpMRI. Considering two different definitions of EPE, any EPE (i.e., focal and established) and established EPE only, the SE and AUC in men with unfavourable intermediate-risk PCa differed from 54.2 and 0.659 to 47.1 and 0.582, respectively [22]. An important and novel finding of this study is that MRI scan quality influences the ability of MRI to predict EPE, and therefore this may, for the first time, also help to further explain differing diagnostic accuracy results achieved in the literature until now. This study clearly shows that better mpMRI scans greatly improve the ability to accurately predict EPE.

Thirteen different scanners were used across our referral network. All scans had at least two functional sequences available for review, and 103/121 scans had DCE sequences. Our results show that the AUC improves with each additional imaging sequence reviewed, including DCE sequences. This finding is supported by the sub-analysis performed by de Rooij et al. within their systematic review, who noted that the SE for EPE improved from 53%, to 62%, to 69% for T2WI, to T2WI + one functional technique, to T2WI + two functional techniques, respectively, with no decrease in SP [10]. Additionally, similar to our results, the incorporation of clinical information is known to improve diagnostic accuracy compared to studies where mpMRI images alone are used [11]. Moreover, only once clinical information and all MRI sequences were available was inter-observer agreement using the Likert score substantial (i.e., weighted Kappa > 0.6) between all radiologists (see Appendix A).

With regards to the additional value of DCE sequences in our study, the administration of gadolinium contrast in high quality scans appears to improve diagnostic accuracy considerably, but not when the scan is of a lesser technical quality (PI-QUAL 1–3) (see Table 3 and Appendix A). Caglic et al. showed that mpMRI was no better than biparametric MRI at detecting EPE, but mpMRI was better for detecting seminal vesicle invasion; however, they did not study the impact of an objective quality assessment of the scan [23]. Interestingly, in our study, in scans where DCE sequences were performed, in the locked sequential analysis, the AUC after review of DCE images was 0.76 compared to 0.69 before for T2WI and DWI images only (see Appendix A). Results such as these, which suggest the value of DCE images for the accurate staging of PCa, raise the question of whether men proceeding to RP will require an additional scan including gadolinium contrast in order to best inform the conduct of their surgery if such an mpMRI scan has not been performed in the first instance. Although artefactual changes seen on MRI after biopsy can compromise interpretation, in our study men who had a repeat scan following biopsy had excellent levels of diagnostic accuracy (AUC of 0.93, Appendix A). Notably, all these scans were a minimum of 63 days since biopsy, all had DCE sequences, and all scans were PI-QUAL score 3 or above. 

mpMRI for EPE prediction is improved when radiologists use scores that convey the relative likelihood of EPE, rather than relying on binary terms such as ‘present’ or ‘absent’ [10]. The value of a risk-stratified ordinal scale to predict the likelihood of EPE is supported by the findings of this study where detection rates of EPE increased with each increment in the score assigned: 0%, 4.3%, 13.9%, 41.4%, and 73.8% for scores 1, 2, 3, 4, and 5, respectively (*p* < 0.0001) (Figure 3 and Appendix A). Risk-based grading systems have been described in the successful prediction of other features of PCa aggressiveness, such as biopsy ISUP score [24]. Indeed, there are several other methods of providing risk-stratified prediction scales for EPE based on mpMRI reported in the literature. These include (but are not limited to) EPE grade [25], ESUR [26], curvilinear contact length (CCL) [27], and tumour capsule length (TCL) [28]. The EPE Grade system has been externally validated with promisingly similar results [29]. Importantly, Park et al. have shown that different methods can perform similarly [30], and there is good evidence to consistently suggest that the incorporation of clinical details alongside mpMRI interpretation (such as in our study) may also improve diagnostic accuracy. In our study, we also collected information on the visual features observed in each scan, including tumour capsule abutment, prostate irregularity, NVB thickening, capsule bulge, and ‘measurable EPE’ in accordance with the 2012 ESUR guidance [26]. The frequency of these visual features according to positive scan (Likert ≥ 3) and positive pathology (EPE) can be viewed in the Appendix A.

Interest in mpMRI has led to its inclusion into multivariable EPE prediction models. A model by Gandaglia et al. has been presented and externally validated [31], but their nomogram does not provide EPE likelihood on a side-specific basis [32], whereas NS decisions are made on a side-by-side basis. The SE for pathological EPE according to mpMRI prediction alone in 333 patients analysed in their series was 25% and provided a limited improvement in the accuracy of their model: an AUC of 0.67 without mpMRI data vs. 0.7 with. Other nomograms, such as those presented by Soeterik et al. and Nyarangi-Dix et al. that include mpMRI EPE information in a side-specific fashion (as in our study), perform better, with AUCs of 0.82 and 0.86, respectively [33,34]. Radiomic platforms using MRI prostate features combined with machine learning and clinical information have already shown a comparable diagnostic performance to radiologists and perform better than clinical nomograms, with AUCs of 0.74, 0.75, and 0.67 for radiomic evaluation, radiologist interpretation, and MSKCC nomogram, respectively [35]. 

If the threshold used to classify EPE prediction is modified to allow a higher SP with the Likert threshold moved to 4 instead of 3, the SE, SP, PPV, and NPV would change to 67.2%, 84.5%, 49.4%, and 92%, respectively (Appendix A). This would potentially facilitate more aggressive NS decision-making, but at the same time more men may incur PSM as a result. Another option when there is concern but uncertainty for EPE is intra-operative margin assessment, such as the NeuroSAFE technique or others [36,37]. However, when teams have evaluated their own performance and EPE is highly likely (for instance, 73.8% in Likert 5 in this series), aggressive ipsilateral NS may not be warranted. 

This work has some limitations. Firstly, our centre benefits from radiologists with high levels of mpMRI expertise. A consideration of paramount importance for any team using mpMRI for EPE prediction to help guide the NS decision is that they must evaluate and understand their own diagnostic performance. Conversely, the MRIs in this study were carried out at a number of community hospitals, which improves the external generalisability of these results. Secondly, though blinded to the final histology, our radiologists reviewed scans retrospectively. Prospective validation studies with a larger pool of contributing radiologists are planned. Thirdly, our cohort is limited to patients undergoing RP and to those included in the NeuroSAFE PROOF study. Thus, the cohort is biased towards younger patients who had good erectile function at baseline. As such, these findings may not readily transfer to patient cohorts such as undiagnosed men or men undergoing other treatments, though the availability of a reference test for pathological EPE is not possible in these men. Fourthly, by excluding from our analysis lobes wherein an intra-prostatic PSM was noted without EPE elsewhere on the ipsilateral side, we may have introduced some bias in our study population, though the absence of a valid reference standard in these lobes prevented their inclusion. Fifthly, this is a retrospective study with a relatively small number of patients and as a consequence of the small numbers, we report broad confidence intervals in relation to the diagnostic performance parameters. In the future, the authors intend to incorporate more participants, centres, and radiologists using previously unseen mpMRI scans to prospectively validate whether the same results can be achieved in external settings but using the same interpretation methods. Moreover, small numbers of participants may contribute to some results which lie outside expected references, for instance the high proportion of men with biopsy ISUP 3 and 4 disease had EPE. 

## 5. Conclusions

Likert score can be used very effectively by expert genitourinary radiologists to rule out EPE on a side-specific basis in men undergoing RARP. The diagnostic accuracy of mpMRI for predicting pathological EPE improves with the availability of additional MRI sequences, sight of clinical information, and better quality mpMRI scans.

## Figures and Tables

**Figure 1 diagnostics-12-01057-f001:**
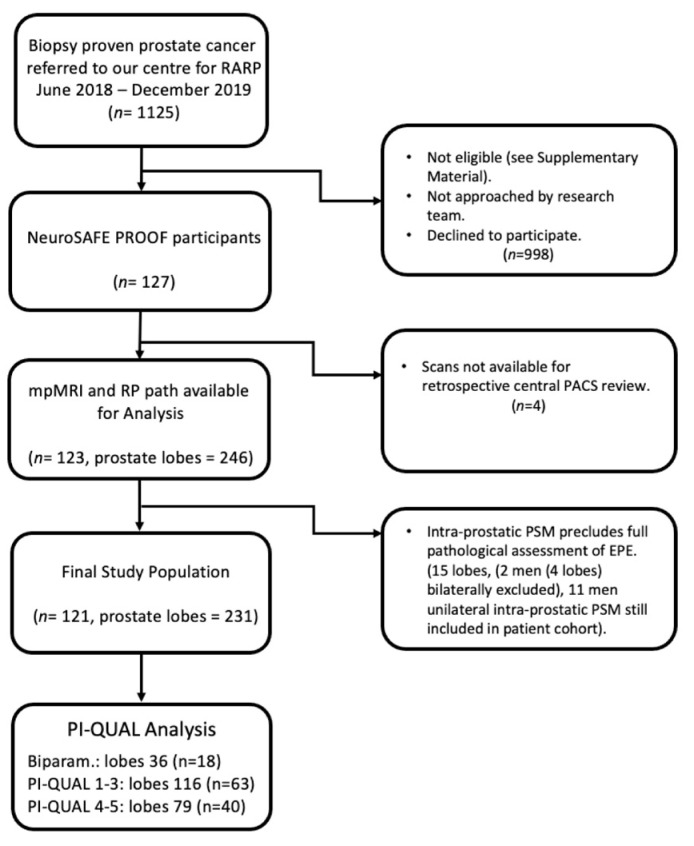
Flow diagram of patient selection for study. *Abbreviations*: RARP, robot-assisted radical prostatectomy; PACS, picture archiving and communication system; RP, radical prostatectomy; PSM, positive surgical margin; EPE, extra-prostatic extension, PI-QUAL, Prostate Imaging Quality score; Biparam., bi-parametric MRI scan.

**Figure 2 diagnostics-12-01057-f002:**
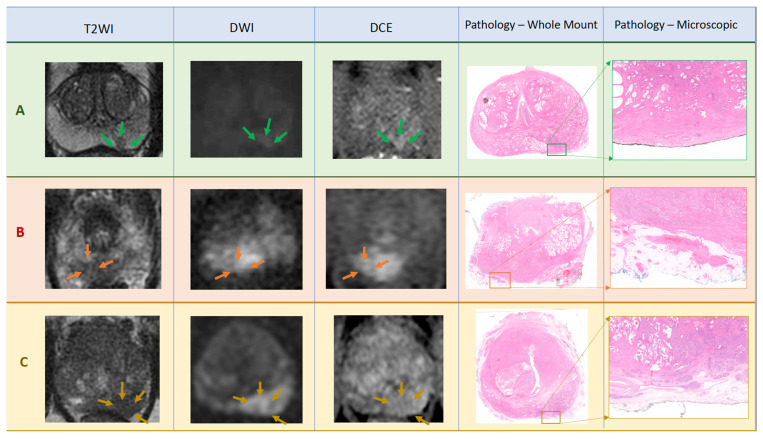
Panel of example mpMRI scan images with corresponding histological RP whole mount images including areas of interest for EPE displayed in further magnification. Patient **A**: Images from mid-gland show Likert score 2 Left-lesion in left peripheral zone posterior medial (green arrows). Corresponding pathology panels show pT2c including disease confined to the gland, and not extending to the inked margin, in the left posterior medial peripheral zone (×1.5). Patient **B**: Images from apex of gland show Likert score 4 Right-lesion in right peripheral zone posterior medial showing irregularity and bulge (orange arrows). Corresponding pathology panels show pT3a including disease extending into the fat for 2 mm but clear of the inked margin (×1.42). Patient **C**: Images from base of gland show Likert score 5 Left-lesion in left peripheral zone posterior medial and posterior lateral demonstrating abutment, irregularity, bulge, and ‘measurable EPE’ (gold arrows). Corresponding pathology panels show pT3a including disease extending circumferentially beyond capsule and into the fat for 8 mm but clear of the inked margin (×1.5). All DWI images have b value of either b1400 or b2000.

**Figure 3 diagnostics-12-01057-f003:**
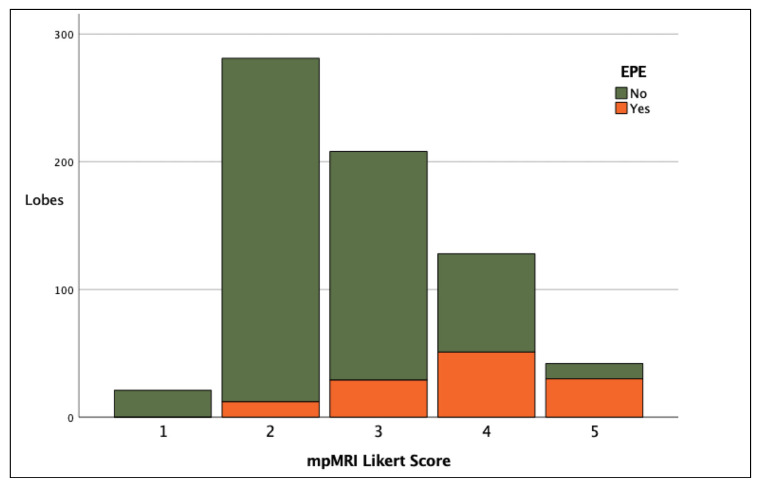
Stacked bar chart showing clinical detection of EPE per lobe according to final Likert score.

**Table 1 diagnostics-12-01057-t001:** Patient, radiological, and pathological characteristics of study cohort.

		All (n = 121)	No EPE (n = 82)	EPE Present (n = 39)	*p* Value †
Patient	Mean age, year ± SD (range)	56.9 ± 7	56.6 ± 7.5	57.82 ± 5.6	0.34
		(40–71)	(40–71)	(48–71)	
	Mean PSA, ng/mL ± SD (range)	8.9 ± 6	7.1 ± 3.8	12.7 ± 7.9	<0.001
		(1.2–35)	(1.2–25)	(4.6–35)	
	Clinical ISUP, n (%)				<0.001
	1	6 (5)	5 (83.3)	1 (16.7)	
	2	94 (77)	72 (76.6)	22 (23.4)	
	3	14 (11.6)	4 (28.6)	10 (71.4)	
	4	6 (5)	1 (16.7)	5 (83.3)	
	5	1 (0.8)	0	1 (100)	
	EAU risk, n (%)				0.013
	Low	0	0	0	
	Int	34 (28.1)	28 (34.1)	6 (15.4)	
	High	87 (71.9)	54 (65.9)	33 (84.6)	
	DRE *, n (%)				<0.001
	T1	53 (22.9)	44 (83)	9 (17)	
	T2	125 (54.1)	111 (89)	14 (11)	
	T3	53 (22.9)	33 (62.3)	20 (37.7)	
mpMRI	Mean time from MRI to operation, days ± SD (range)	108 ± 68.8	114 ± 73.1	98 ± 58.3	0.23
		(2–355)	(2–355)	(6–292)	
	Mean prostate volume, cc ± (range)	36.6 ± 15.0	36.3 ± 15.1	37.2 ± 14.8	0.75
		(12–86)	(13–86)	(14–80)	
	Tumour side, n (%)				0.58
	Right	35 (28.9)	23 (65.7)	12 (34.3)	
	Left	22 (18.2)	16 (72.7)	6 (27.3)	
	Both	61 (50.4)	42 (68.9)	19 (31.1)	
	No visible lesion	3 (2.5)	1 (33.3)	2 (66.6)	
	Tumour position, n (%)				0.39
	Posterior				
	Anterior	97 (80.2)	68 (70.1)	29 (29.9)	
	Both	8 (6.6)	4 (50)	4 (50)	
	No visible lesion	13 (10.7)	9 (69.2)	4 (31.8)	
		3 (2.5)	1 (33.3)	2 (66.6)	
RP specimen	Mean prostate weight, g ± SD (range)	43.8 ± 13.9	44.6 ± 14.4	42.1 ± 12.7	0.36
		(15–89)	(22–89)	(15–74)	
	Mean tumour volume, mls ± SD (range)	4.3 ± 3.6	3.4 ± 2.8	6.3 ± 4.5	<0.001
		(0.25–22.3)	(0.25–12.9)	(1.2–22.30	
	Pathological ISUP, n (%)				0.003
	1				
	2	2 (1.7)	2 (100)	0	
	3	92 (75.4)	69 (75)	23 (25)	
	4	23 (18.9)	10 (43.5)	13 (56.5)	
	5	1 (0.8)	1 (100)	0	
		3 (2.5)	0	3 (100)	
	Pathological stage, n (%)				n/a
	2a-b				
	2c	7 (5.8)	7	0	
	3a	75 (62)	75	0	
	3b	32 (26.4)	0	32	
		7 (5.8)	0	7	

*Abbreviations*: ISUP, International Society of Urological Pathology; PSA, prostate specific antigen; SD, standard deviation; EAU, European Association of Urology. * Denominator given as lobes included in the study (n = 231). † Statistical tests used; independent t test where mean is provided, chi-squared test where data is categorical and provided as proportions.

**Table 2 diagnostics-12-01057-t002:** Per prostate lobe analysis of sensitivity (SE), specificity (SP), positive predictive value (PPV), negative predictive value (NPV), and area under the curve (AUC) by individual radiologist and combined where Likert score 3≥ was positive scan for pathological EPE.

	Reader 1	Reader 2	Reader 3	Readers Combined
SE *	88.4	92.5	90.5	90.4
	(74.9–96)	(79.6–98.4)	(77.4–97.3)	(83.8–94.9)
SP *	61.7	39.5	55.5	52.3
	(54.4–68.7)	(32.4–46.9)	(48–62.9)	(48–56.5)
PPV *	34.6	24.8	31.9	29.9
	(25.7–44.2)	(18.1–32.6)	(23.7–41.1)	(25.3–34.8)
NPV *	95.9	96.1	96.2	96
	(90.6–98.6)	(88.9–99.2)	(90.5–99)	(93.2–97.9)
AUC	0.84	0.77	0.83	0.82
	(0.77–0.92)	(0.68–0.86)	(0.76, 0.9)	(0.77–0.86)

* Values in parentheses are 95% confidence intervals.

**Table 3 diagnostics-12-01057-t003:** Diagnostic performance of MRI for prediction of EPE final Likert score ≥ 3 for biparametric MRI scans and mpMRI PI-QUAL scores of 1–3 and 4–5.

	Biparametric Scan (n = 18)	PI-QUAL 1–3 (n = 63)	PI-QUAL 4–5 (n = 40)
SE	80 (59.3–93.2)	89.1 (78.8–95.5)	100 (90.3–100)
SP	48.4 (37.9–59)	49.8 (43.7–56)	57 (49.7–64.1)
NPV	90 (78.2–96.7)	95 (90–98)	100 (96.7–100)
PPV	29.4 (19–41.7)	29.8 (23.5–36.9)	30.3 (22.2–39.4)
AUC	0.76 (0.64–0.88)	0.78 (0.72–0.84)	0.92 (0.88–0.96)

Values in parentheses are 95% confidence intervals.

## Data Availability

All datasets generated for this study are included in the manuscript. E.D. had full access to all the data in the study and takes responsibility for the integrity of the data and the accuracy of the data analysis.

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
