# Peer review of "Negative mpMRI Rules Out Extra-Prostatic Extension in Prostate Cancer before Robot-Assisted Radical Prostatectomy"

_diagnostics, 2022, doi:10.3390/diagnostics12051057_

Round 1

Reviewer 1 Report

In this study, Dinneen et al. evaluated the negative mpMRI rules out extra-prostatic extension in prostate cancer before robot-assisted radical prostatectomy. The idea is of interest and fits the journal scope, but some major points should be addressed as follows:

1. More literature review should be added in the "Introduction" in terms of related works on this problem.

2. It is important to have an external validation data to evaluate the performance of model on unseen data.

3. When comparing the performance among readers, it is necessary to have some statistical tests to see the significant differences.

4. ROC curves should be plotted in addition to AUC.

5. I'm not sure whether can we compare the performance with previously published works on the same problem in terms of measurement metrics.

6. Metrics (i.e., Sens, Spec, AUC, ...) have been used in previous studies such as PMID: 33735760, PMID: 34502160. Thus, the authors are suggested to refer to more works in this description to attract a broader readership.

7. Table 1 should have some statistical tests and p-values.

Author Response

In this study, Dinneen et al. evaluated the negative mpMRI rules out extra-prostatic extension in prostate cancer before robot-assisted radical prostatectomy. The idea is of interest and fits the journal scope, but some major points should be addressed as follows:

  1. More literature review should be added in the "Introduction" in terms of related works on this problem.

Thank you.  Yes, entirely agreed that the introduction does not do justice to the size and interest in the clinical problem.  I have re-worded the second paragraph and promoted reference to the more recent review paper related to MRI for PCa staging.  (See below).  I hope this will provide more by way of outlining the need for more work in this area.

‘Since this review, although use of mpMRI has become vastly more prominent in the PCa diagnostic pathway around the world, a recent review by Abrams-Pompe et al clearly demonstrates the ongoing challenges of using this modality for staging purposes (10).  In their review (similar but more pronounced to the aforementioned 2016 systematic review) they continued to demonstrate enormous variation in the diagnostic performance with SE ranging from 0-100%, which the authors attribute to differing sample size, event rates, reader experience levels, study design, and heterogeneous non-standardized reporting of mpMRI.  Interestingly, although clearly a critical part of the performance and interpretation of mpMRI, technical scan quality has never been assessed as a contributing factor to the accuracy of mpMRI for PCa staging perhaps because until recently there has been little attention to grading mechanisms for scan quality (11-13).  Thus, although mpMRI is now routinely used for pre-operative PCa staging, variation in its performance is huge and, as such, its role in surgical planning remains controversial.’

  1. It is important to have an external validation data to evaluate the performance of model on unseen data.

Thank you for this comment.  The use of the Likert score to predict EPE is not a multivariable model, and as such there is no plan currently to apply it to an unseen data set for external validation as such.  However, we very much agree with the reviewer that the generalisability of this method (the Likert scoring system 1-5) is a very important aspect to this work.  As such, our next step, which is already underway is to apply this method in a similar fashion prospectively and in larger numbers than reported in the current study.  This will be the next important step in evaluating the performance of Likert for this purpose in unseen data and we look forward to reporting the findings of this work. 

This ambition is referred to now in the manuscript in the limitations section of the Discussion ‘Prospective validation studies with a larger pool of contributing radiologists are planned.’

  1. When comparing the performance among readers, it is necessary to have some statistical tests to see the significant differences.

Thank you for this valid comment.  We have now conducted and included an analysis including inter-observer agreement and presented this using weighted kappa coefficient.  In order to comply with editorial instructions I have provided this information in a Table in the revised Supplementary Material (Table S9) but have referred readers to this results table in the Results section and the Discussion section. 

Now included in Supplementary Material:

‘Table S9  Inter-reader agreement (κw) for Likert based mpMRI assessment of EPE.

Readers

T2WI only

+DWI

+DCE

+PSA

Final Read†

1 & 2

0.66

(0.57 - 0.74)

0.63

(0.54 - 0.72)

0.6

(0.51 - 0.7)

0.64

(0.56 - 0.72)

0.62

(0.54 - 0.7)

1 & 3

0. 64

(0.48 - 0.65)

0.57

(0.48 - 0.67)

0.58

(0.48 - 0.67)

0.6

(0.51 - 0.69)

0.64

(0.56 - 0.72)

2 & 3

0.56

(0.48 - 0.65)

0.54

(0.45 - 0.63)

0.55

(0.45 - 0.65)

0.59

(0.51 - 0.67)

0.64

(0.57 - 0.71)

†Including information from the prostate biopsy. 

Data are weighted Kappa coefficient (κw) statistics and data in parentheses are 95% confidence intervals.  Traditional conventions; Kappa (K) <0 indicates no agreement; K= 0-0.2, slight agreement; K = 0.21-0.4, fair agreement; K = 0.41-6, moderate agreement; K = 0.61-0.8, substantial agreement; K = 0.81-1, almost perfect agreement.  All analyses were performed using SPSS software (version 27, IBM).’

Now included in Results Section:

‘Inter-observer agreement was determined using weighted Kappa coefficient and is provided in Supplementary Material Table S9 showing substantial agreement between all readers on the final Likert score for detection of EPE.’

Now included in Discussion Section:

‘Moreover, only once clinical information and all MRI sequences were available was inter-observer agreement using the Likert score substantial (i.e. weighted Kappa >0.6) between all radiologists (see Supplementary Table S9).’

  1. ROC curves should be plotted in addition to AUC.

Thank you.  ROC curves were provided originally in the Supplementary Material.  Please see Figure S11.

  1. I'm not sure whether can we compare the performance with previously published works on the same problem in terms of measurement metrics.

Thank you for this comment.  The authors certainly agree that given the heterogeneity in the studies previously published on this question, comparison of measurement metrics is challenging.  Indeed, in total agreement with the reviewers comment, the authors try to explicitly highlight how challenging this comparison may be by investigating reasons for the broad variation in reported performance in the Discussion (see fifth paragraph of Discussion).  However, previous authors on the same topic have sought to compare their work to the performance of others, and we also feel that the comparison of our work to other authors in this field is similarly justified by the fact that similar efforts have been made by the authors of previous systematic review including the performance of a meta-analysis.

Reviewer 2 Report

Authors should be congratulated for their great contribution to the challenging topic. All future perspectives should lead to improving prostate cancer detection and reducing investigations number. Moreover, the future purpose is to create new and better algorithms to properly manage early-stage PCa patients, avoiding overdiagnosis and overtreatment. Furthermore, a detailed pre-operative assessment of PCa is imperative both for the surgeons and for PCa patients’ expectations. The manuscript is well written, the population is well-enrolled, tables and figures are clear but there are several points that warrant a comment:

  1. Why did Authors not include in the evaluation of mpMRI the Apparent diffusion coefficient (ADC) reconstruction?
  2. Authors should read this novel paper (DOI:1007/s00330-021-08332-8) on the value of mpMRI for the prediction of PCa aggressiveness.
  3. How do Authors explain the high percentage of EPE in ISUP 3 and 4 (71,4% and 83.3%, respectively) ?? A previous misinterpretation of mpMRI or a biological behavior and upstaging of tumor at RARP compared to biopsy specimen?
  4. How this technique differs from radiomics to localize the PCa? Authors should read this novel paper about the role of radiomics in the localization of Clinically significant PCa. (31153556)

Author Response

Authors should be congratulated for their great contribution to the challenging topic. All future perspectives should lead to improving prostate cancer detection and reducing investigations number. Moreover, the future purpose is to create new and better algorithms to properly manage early-stage PCa patients, avoiding overdiagnosis and overtreatment. Furthermore, a detailed pre-operative assessment of PCa is imperative both for the surgeons and for PCa patients’ expectations. The manuscript is well written, the population is well-enrolled, tables and figures are clear but there are several points that warrant a comment:

Thank you very much.  We appreciate this feedback and we entirely agree with the ambitions also outlined by this reviewer.

  1. Why did Authors not include in the evaluation of mpMRI the Apparent diffusion coefficient (ADC) reconstruction?

Thank you for this helpful point and the opportunity to improve the clarity of the methods.  The three radiologists were able to view ADC reconstruction as part of the DWI sequences.  Though lesion ADC values were not specifically calculated and used as a single ‘visual characteristic’ such as those listed in Supplementary Tables S11 and S12, the ADC images were cogntively incorporated into the Likert scores assigned by the radiologists as part of viewing the DWI sequences in the locked sequential reads and subsequent reads.

The Methods section has been updated to ensure this is now more clear for the reader:

‘Images were reviewed and reported in locked sequential manner such that T2WI was reported first, then DWI (including apparent diffusion coefficient (ADC) reconstruction maps), then DCE, then blood serum prostate specific antigen (PSA) level was provided, and finally prostate biopsy information was provided.’

2.  Authors should read this novel paper (DOI:1007/s00330-021-08332-8) on the value of mpMRI for the prediction of PCa aggressiveness.

Thank you for suggesting this very interesting paper.  There appears to be clear corollaries between the 3-point ordinal MRI grading group system (mG1 to mG3) described by Boschheidgen and colleagues in this paper that predicts biopsy ISUP aggressiveness and the Likert score used to predict likelihood of EPE (i.e. also a feature of adavances PCa aggressiveness) in our study.  I have included reference to this paper now in the discussion:

‘Risk based grading systems have been described in the successful the prediction other features PCa aggressiveness, such as biopsy ISUP score.’

3. How do Authors explain the high percentage of EPE in ISUP 3 and 4 (71,4% and 83.3%, respectively) ?? A previous misinterpretation of mpMRI or a biological behavior and upstaging of tumor at RARP compared to biopsy specimen?

Thank you for this insightful comment.  We agree that the rates of EPE in our series appear to be higher when analysed as per biopsy Gleason grade 4+3 and 4+4 than typically recognised in the literature.  EPE columns in Table 1 refer to pathological EPE on final specimen.  On the whole, our explanation is that the small numbers of men in this series with ISUP 3 (n=14) and ISUP 4 (n=6) may be responsible for variations in proportions that would not be evident if the numbers were larger.  In ISUP 2 (i.e. Gleason 3+4) there are 94 men and 23.4% of them (n=22) had EPE, which is much more in line with the reported literature.  Thank you for pointing this out.  We comment on the relatively high proportion of EPE in patients with ISUP 3 and 4 in the limitations section of the discussion:

‘Fifthly, this is a retrospective study with a relatively small number of patients.  Conducting the same methods and achieving the same high-performance results prospectively at multiple centres will contribute to the promise and the importance of the findings reported here.  Moreover, small numbers of participants may contribute to some results which lie outside expected references, for instance the high proportion of men with biopsy ISUP 3 and 4 disease had EPE.’

4. How this technique differs from radiomics to localize the PCa? Authors should read this novel paper about the role of radiomics in the localization of Clinically significant PCa. (31153556)

Thank you.  The authors agree.   Have included the following sentence in the discussion and referred to the suggested paper.

‘Radiomic platforms using MRI prostate features combined with machine learning and clinical information, have already shown comparable diagnostic performance to radiologists and perform better than clinical normograms (35).’

Reviewer 3 Report

This study aims to investigate how multi-parametric MRI (mpMRI) could accurately predict PCa extra-prostatic extension (EPE) on a side-specific basis using a risk-stratified 5-point Likert scale. The authors included 124 men who underwent robot-assisted RP (RARP) between June 2018 and December 2019.

They concluded that MRI can be used effectively by genitourinary radiologists to rule out EPE and help inform surgical planning for men undergoing RARP.

I believe that it is worthy of publication after major revision.

MAJOR COMMENTS

  • I suggest providing a more detailed presentation of the use of mpMRI in prostate cancer. In this regard, this interesting paper (doi: 10.23736/S0393-2249.20.03688-7; PMID 32182231) shows the low sensitivity of mpMRI during staging for focal (microscopic) extra-prostatic extension (EPE). In addition , it will be for the benefit of the reader if the author comment on the concordance rates regarding MRI with the definitive histologic report of prostatectomy specimen (doi: 10.1007/s00261-020-02798-8; PMID 33048224).
  • Do the authors believe in the potential of radiomics-powered Machine Learning for the detection of EPE in PCa? (please discuss doi: 3390/ijms22189971 PMID PMC8465891)
  • The authors should better highlight that the differences in mpMRI performance as a predictor of EPE might reside in the differences in the definition of EPE.

Author Response

This study aims to investigate how multi-parametric MRI (mpMRI) could accurately predict PCa extra-prostatic extension (EPE) on a side-specific basis using a risk-stratified 5-point Likert scale. The authors included 124 men who underwent robot-assisted RP (RARP) between June 2018 and December 2019.

They concluded that MRI can be used effectively by genitourinary radiologists to rule out EPE and help inform surgical planning for men undergoing RARP.

I believe that it is worthy of publication after major revision.

Thank you.  Please see specific comments and revisions underneath each point.

MAJOR COMMENTS

  • I suggest providing a more detailed presentation of the use of mpMRI in prostate cancer. In this regard, this interesting paper (doi: 10.23736/S0393-2249.20.03688-7; PMID 32182231) shows the low sensitivity of mpMRI during staging for focal (microscopic) extra-prostatic extension (EPE).

Thank you for suggesting this nice and helpful paper, which the authors had not come across previously.  It has clear corrolaries to the present study.  There are several ways in which this paper might be referred to in relation to the present study.    For instance, the risk profile of PCa and its relation to  accuracy of mpMRI for the perdiction of EPE is clearly of interest.  We have added the following sentence including citation in the Discussion section:

Additionally, different patient populations can have a marked effect on diagnostics accuracy.  For instance, Falagario et al demonstated AUC for EPE prediction using mpMRI of 0.73 for patients with high-risk PCa compared to 0.52 for patietns with low or very low-risk disease (21).  In the present study, 71.9% of patients had EAU high-risk PCa, which may help to also explain strong diagnostic accuracy results (AUC 0.82).   

  • In addition , it will be for the benefit of the reader if the author comment on the concordance rates regarding MRI with the definitive histologic report of prostatectomy specimen (doi: 10.1007/s00261-020-02798-8; PMID 33048224).

Thank you.  The suggested reference is a very good example of the accurate use of mpMRI that has developed for detection of disease within the gland.  We agree that inclusion of this is warranted in the introduction as it helps sets the scene for the levels of accuracy that mpMRI should aim to achieve for EPE.  We have now included the following sentence in the introduction:

It is well recognised that prostate mpMRI has good concordance with PCa on histopathological evaluation of the final RP specimen for lesions that are within the gland (9).

Thank you.

  • Do the authors believe in the potential of radiomics-powered Machine Learning for the detection of EPE in PCa? (please discuss doi: 3390/ijms22189971 PMID PMC8465891)

Thank you.  The authors agree.   Have included the following sentence in the discussion and referred to an interesting manuscript that uses MRI radiomics to assess EPE in prostate cancer (PMID 31153556).

‘Radiomic platforms using MRI prostate features combined with machine learning and clinical infromation, have already shown comparable diagnostic performance to radiologists and perform better than clinical normograms (35).’

  • The authors should better highlight that the differences in mpMRI performance as a predictor of EPE might reside in the differences in the definition of EPE.

Thank you.  Yes, this is an entirely valid point.  We have now referred to this possible contributory factor in A) the introduciton:

In their review they continued to demonstrate enormous variation in the diagnostic performance with SE ranging from 0-100%, which the authors attribute to differing sample size, event rates, reader experience levels, study design, heterogeneous non-standardized reporting of mpMRI, and even different definitions of histopathological EPE.

And b) the discussion:

Possible explanations for this broad variation in the accurate prediction of EPE published in the literature include differences in study design, measurement metrics, definitions of EPE, and clinical practice. 

We have also included the comment:

‘The study by Falagario and colleagues may also demonstrate the importance of the definition of EPE on the diagnostic performance of mpMRI.  Considering 2 different definitions of EPE, any EPE (i.e. focal and established) and established EPE only, the SE and AUC in men with unfavourable intermediate-risk PCa  differed from 54.2 and 0.659 to 47.1 and 0.582, respectively (22).’

Round 2

Reviewer 1 Report

Some comments have not been addressed well such as:

- The authors skipped my previous comments #6, #7 without any response and clarification.

- It is important to have external validation data to evaluate the performance of model on unseen data.

- CI ranges were too high.

- The author should compare the predictive performance to previously published works on the same problem/data.

Author Response

Please find further responses to Reviewer 1 comments in red below:

Some comments have not been addressed well such as:

- The authors skipped my previous comments #6, #7 without any response and clarification.

Please accept our apologies that we did not answer these with sufficient clarity and precision.

With regards to comment 6, ‘Metrics (i.e., Sens, Spec, AUC, ...) have been used in previous studies such as PMID: 33735760, PMID: 34502160. Thus, the authors are suggested to refer to more works in this description to attract a broader readership.’ 

The authors agree with the reviewer that the use of radiomics in MRI prostate may be a very interesting and important innovation.  We have now included the following sentence and reference, which we hope will demonstrate the potential future value of radiomics particularly in relation to MRI prostate:

‘Radiomic platforms using MRI prostate features combined with machine learning and clinical information, have already shown comparable diagnostic performance to radiologists and perform better than clinical nomograms; AUC 0.74, 0.75 and 0.67 for radiomic evaluation, radiologist interpretation and MSKCC nomogram, respectively (35).’

We decided not refer to the 2 articles specifically mentioned by Reviewer 1 as these refer to classification of transcriptome sub-types in glioblastoma (PMID: 33735760) and prediction of non-small cell lung cancer (PMID: 34502160).  Whilst these are interesting and important articles with diagnostic accuracy values provided, the availability of contemporary research on MRI prostate radiomics for the accurate prediction of EPE (new reference 35, Losnegard et al) also with diagnostics accuracy values provided, the authorts felt this reference was more directly of relevance to urologists and uro-radiologists.

With regards to comment 7, ‘Table 1 should have some statistical tests and p-values.’  As well as including the p values in the table as suggested by Reviewer 1, we have also now added the following sentence to the Results section:

‘Preoperative PSA, clinical (i.e. biopsy) ISUP score, EAU risk category, DRE, RP specimen tumour volume, and RP ISUP score differed significantly between the groups with EPE and without EPE (all p<0.05).’   

PSA, ISUP score, EAU risk category, DRE, tumour volume, and RP ISUP are all well-recognised risk factors for more aggressive PCa pathology / phenotype, and therefore EPE.  As such, the authors have not discussed these findings extensively in the Discussion as Table 1 is intended more to tabulate the demographics and disease characteristics of the participants included in the study.

- It is important to have external validation data to evaluate the performance of model on unseen data.

Thank you.  The authors agree that under ideal circumstances there would be an external data set comprising of scans not previously seen by the radiologists for an external validation of the interpretation techniques used (Likert ordinal score 1-5).  However, we cannot include this work in this manuscript as we have not yet performed this work.  We are enthused by the results we present in this paper, and as such, future evaluation using more scans, external data/scan sources and additional radiologists to validate this approach are warranted.  Therefore, we have included the following sentence in the Limitations section of the Discussion to highlight this, and to point to future work addressing the same matter:

In the future, the authors intend to incorporate more participants, centres, and radiologists and use previously unseen mpMRI scans to prospectively validate whether the same results can be achieved in external settings using the same interpretation methods.’

- CI ranges were too high.

Thank you.  Though it is not clear which area of the results the reviewer is referring to, the authors certainly agree that the broad CI’s are a limitation of the study related to the relatively small numbers of participants.  We have now included the following sentence in the list of limitations of the study to highlight the broad CI’s and the explanation for this limitation of the research:

Fifthly, this is a retrospective study with a relatively small number of patients and as a consequence of the small numbers, we report broad confidence intervals in relation to the diagnostic performance parameters.’ 

- The author should compare the predictive performance to previously published works on the same problem/data.

Thank you.  There is a large amount of work already performed on this question in the published literature.  In paragraph 4 of the Discussion the authors have tried to refer to some of this work and to compare our results to this work.  The following sentence highlights the 2 existing reviews (one of which is a systematic review) on this question and compares our results them findings of these reviews:

‘These results [of our study], high SE and lower SP, are at odds with the findings of the systematic review and meta-analysis of MRI for local PCa staging performed by de Rooij et al in 2016.  This review included 75 studies and demonstrated high SP (91%) with poor and heterogeneous SE (51%) (10).  Since this review huge interest has continued in this topic as highlighted by the review by Abrams-Pompe et al who include a further 48 studies that have been published with significant ongoing variation in diagnostic accuracy reported (11)

Additionally, in the remainder of the paragraph, the authors refer to 3 other original research articles, referring directly to the sensitivity and specificity performance characteristics.  In the rest of the discussion, the authors believe that several other research papers on the same question are referred to and compared to the performance of Likert score in the prediction of EPE on mpMRI in this research.  The authors agree that there are a number of other research articles that have not been directly referred to in relation to this work, however, the abundance of published work on this question would make this not possible in a research article such as ours.

Reviewer 3 Report

Tra authors answered all suggestions and comments.

Author Response

Thank you.

Round 3

Reviewer 1 Report

My previous comments have been addressed.

Author Response

Thank you for your kind reply.